# First-line modified FOLFOX plus/minus nivolumab and Ipilimumab or FLOT plus nivolumab in advanced gastroesophageal adenocarcinoma: a phase II multi-cohort IKF-AIO-MOONLIGHT trial

Sylvie Lorenzen [1] ✉, Thorsten Oliver Goetze [2], Peter C. Thuss-Patience[3], Jorge Riera-Knorrenschild[4], Eray Goekkurt[5], Tobias Nicolaas Dechow[6], Thomas Jens Ettrich [7], Ralf Dieter Hofheinz[8], Kim Barbara Luley[9], Daniel Pink[10], Udo Lindig[11], Gunnar Folprecht[12], Gunter Schuch[13], Michael Bitzer [14], Volker Heinemann[15], Stefan Angermeier[16], Claus Bolling[17], Maria Loose[18], Sabine Junge[18], Claudia Pauligk[18] & Salah-Eddin Al-Batran[2]

This multi-cohort study evaluates whether dual immune checkpoint inhibition with nivolumab and ipilimumab in parallel or sequentially, or triplet chemotherapy with nivolumab can enhance efficacy as first-line therapy for advanced or metastatic gastroesophageal adenocarcinoma. Patients are randomized 1:1 to Arm A (mFOLFOX + nivolumab 240 mg every two weeks + ipilimumab 1 mg/kg every six weeks in parallel or Arm B (mFOLFOX). Subsequently, patients are randomized 1:2 to Arm A1 (identical to Arm A) or Arm A2 (three cycles of mFOLFOX followed by nivolumab + ipilimumab). In Arm C all patients receive FLOT + nivolumab. Primary endpoint is progression-free survival. Secondary endpoints include overall survival and objective response rate. Median progression-free survival (months) is: Arm A/A1, 5.8; Arm A2, 4.0; Arm B, 6.6 and Arm C, 7.0.Toxicity is manageable across all arms, with higher rates in arms receiving dual checkpoint inhibition. Here we show that chemotherapy with dual checkpoint inhibition in parallel increases toxicity without improving efficacy, while FOLFOX followed by immunotherapy is insufficient. FLOT with nivolumab appears feasible and shows promising efficacy. ClinicalTrials.gov number NCT03647969

Over the past decade, first-line treatment with PD-1 or PD-L1 blockade combined with standard chemotherapy regimen as a combination of a platinum drug (cisplatin or oxaliplatin) and a fluoropyrimidine (fluorouracil, capecitabine), has transformed the treatment landscape for HER2-negative metastatic gastroesophageal adenocarcinoma (GEA) (1; 2).

CheckMate-649 was the first large-scale phase 3 randomized trial to evaluate nivolumab plus chemotherapy (nivo+CT) versus chemotherapy (CT) alone in patients with metastatic HER2- negative GEA and demonstrated a clinically meaningful improvement in overall survival in patients with PD-L1 combined positive score ≥5

**Table 1 | Baseline characteristics of the intention-to-treat population according to treatment arm**

| | Arm A/A1 N = 90 | Arm A2 N = 60 | Arm B N = 60 | Arm C N = 52 |
|---|---|---|---|---|
| **Age (years)** | | | | |
| Median | 58 (27–84) | 63 (33–82) | 63 (33–82) | 61 (29–87) |
| **Sex** | | | | |
| Male | 61 (68%) | 39 (65%) | 39 (65%) | 34 (65%) |
| Female | 29 (32%) | 21 (35%) | 21 (35%) | 18 (35%) |
| **ECOG** | | | | |
| 0 | 54 (60%) | 39 (65%) | 32 (53%) | 24 (46%) |
| 1 | 35 (40%) | 21 (35%) | 28 (47%) | 28 (54%) |
| **Location primary** | | | | |
| GEJ I-III | 55 (61%) | 29 (48%) | 29 (48%) | 34 (65%) |
| Stomach, corpus or antrum | 34 (38%) | 31 (52%) | 30 (50%) | 18 (35%) |
| Missing | 1 (1%) | - | 1 (2%) | - |
| **Prior resection of primary tumor** | | | | |
| Yes | 33 (37%) | 22 (37%) | 21 (35%) | 11 (21%) |
| No | 57 (63%) | 38 (63%) | 39 (65%) | 41 (79%) |
| **Lauren's type** | | | | |
| Diffuse | 25 (28%) | 14 (23%) | 10 (17%) | 18 (35%) |
| Intestinal | 27 (30%) | 17 (28%) | 25 (42%) | 17 (33%) |
| Mixed | 5 (6%) | 7 (12%) | 4 (7%) | 4 (8%) |
| Not evaluable/Missing | 33 (37%) | 22 (37%) | 21 (35%) | 13 (25%) |
| **Signet-ring cells** | | | | |
| With | 24 (27%) | 24 (40%) | 21 (35%) | 20 (39%) |
| Without | 57 (63%) | 35 (58%) | 33 (55%) | 32 (62%) |
| Missing | 9 (10%) | 1 (2%) | 6 (10%) | - |
| **Grading according to WHO** | | | | |
| G1 | 1 (1%) | 3 (5%) | 1 (2%) | 1 (2%) |
| G2 | 24 (27%) | 20 (33%) | 24 (40%) | 15 (29%) |
| G3 | 60 (67%) | 33 (55%) | 33 (55%) | 35 (67%) |
| Missing | 5 (6%) | 4 (7%) | 2 (3%) | 1 (2%) |
| **PD-L1 combined positive score (CPS)** | | | | |
| CPS < 1 | 36 (40%) | 18 (30%) | 24 (40%) | 13 (25%) |
| CPS ≥ 1 | 42 (47%) | 26 (43%) | 23 (38%) | 34 (65%) |
| missing | 12 (13%) | 16 (27%) | 13 (22%) | 5 (10%) |

Data are median (range) or n (%). Percentages might not add up to 100 because of rounding.

(HR 0.70) and all randomized patients (HR 0.79) compared to CT alone, which led to FDA and EMA approval of nivo+CT in over 50 countries as the new standard of care.

Considerable effort is focused on how to best leverage immune checkpoint therapy. One strategy to enhance the efficacy of anti-PD-(L)1 therapy is to combine it with the cytotoxic T-lymphocyte-associated protein (CTLA-4) immune checkpoint inhibitor ipilimumab (ipi), given that the two inhibitors impact the immune system through two independent, and possibly complementary, mechanisms of action.[1,2]

Nivolumab and ipilimumab have been proven to be effective in multiple solid tumors,[3–6] and demonstrated clinically meaningful antitumor activity with manageable safety profile in heavily pretreated patients with GEA.[7] However in first-line GEA the ipilimumab/nivolumab treatment arm did not improve outcome versus chemotherapy alone and was even closed early, owing to increased rate of adverse events and early deaths.[8] At the time Moonlight trial was designed, it was known, that treatment of metastatic melanoma with combination of anti-CTLA4 and anti-PD-1 therapy resulted in increased response rates and progression-free survival compared to single agent immunotherapy.[9] In metastatic esophagogastric cancer, combined immune checkpoint blockade may further improve the efficacy of single-agent anti−PD-1

therapy by avoiding tumor immune escape through synergistic T-cell antitumor activity.[10,11] Until now, the immunologic consequences of adding the CTLA-4 checkpoint inhibitor ipilimumab to combined nivolumab plus chemotherapy for patients with treatment naïve HER2 negative GEA have not been investigated.

Previous studies have shown that chemotherapy elicits antitumor effects through the immune system, which might result in increased immunotherapy activity.[12,13]

Given the known toxicities of classic chemotherapy regimens, we hypothesized that three cycles of induction chemotherapy followed by dual immunotherapy would provide early disease control while building on the durable survival benefit provided by nivolumab and ipilimumab, and minimize the side-effects that are associated with a full course of continuous chemotherapy. We therefore aimed to investigate 3 cycles induction chemotherapy with FOLFOX followed by nivolumab plus ipilimumab with optional FOLFOX re-induction versus continuous chemotherapy with nivolumab and ipilimumab.

Albeit nivolumab and pembrolizumab added to doublet chemotherapy prolong OS and PFS, long-term outcome for most patients remains poor. Even more, when administered as monotherapy, there is an increased early mortality with crossing survival curves with the checkpoint inhibitors as compared with chemotherapy.[14–16] This is most likely explained by the fact that patients need time to establish antitumoral immunity, while some patients experience early disease progression and death. This provides a rationale to test whether the intensification of chemotherapy using a triplet (FLOT) instead of the doublet, combined with only nivolumab would be more effective.

Here we report the results of the multi-cohort phase 2 MOON-LIGHT trial (NCT03647969) with the goal to expedite the investigation of novel immunotherapy-based strategies in the metastatic first-line setting of Her2-negative GEA and evaluate whether (a) dual immune checkpoint inhibition with nivolumab (nivo) and ipilimumab (ipi), administered in parallel or sequentially, or (b) triplet chemotherapy combined with immune checkpoint monotherapy can enhance clinical outcomes for GEA.

## Results
### Patients and treatment
Between November 2018 and February 2022, 281 patients were screened. Overall, 262 patients were either randomized into Arm A (n = 60), A1 (n = 30), A2 (n = 60) and B (n = 60) or enrolled in Arm C (n = 52) in 27 trial sites in Germany. Since the only difference between Arm A and A1 was the study period in which the patients were enrolled, and baseline characteristics between the two arms did not reveal any significant differences, the data of both arms were pooled for all analyses (Arm A/A1 n = 90).

The median age ranged between 58 and 63 years between all arms and around two-third of the patients were male. In Arm A/A1 and Arm C the amount of patients with adenocarcinoma of the gastroesophageal junction (GEJ) was higher than of stomach, whereas in Arm A2 and B patients with stomach adenocarcinoma were represented in the same number as patients with GEJ. Overall, only one-third of the patients received prior resection of the primary tumor. Overall PD-L1 expression (CPS ≥ 1) was comparable between Arms A/A1, A2 and B, ranging between 38% and 47% and with 60% higher in Arm C (Table 1)

### Doublet chemotherapy plus IO (Arm A/A1, A2) and chemotherapy alone (Arm B)
The median treatment duration was 5.1 months, 3.3 months and 4.4 months in Arm A/A1, A2 and B, respectively, with 5 (6%), 3 (5%) and 1 (2%) patients completing the protocol-defined maximum treatment duration of 24 months. Disease progression was the major reason of premature treatment discontinuation in all three arms, and 5–7% of the patients discontinued treatment due to unacceptable toxicity (Fig. 1). In Arm A2, 8 patients (13%) discontinued treatment

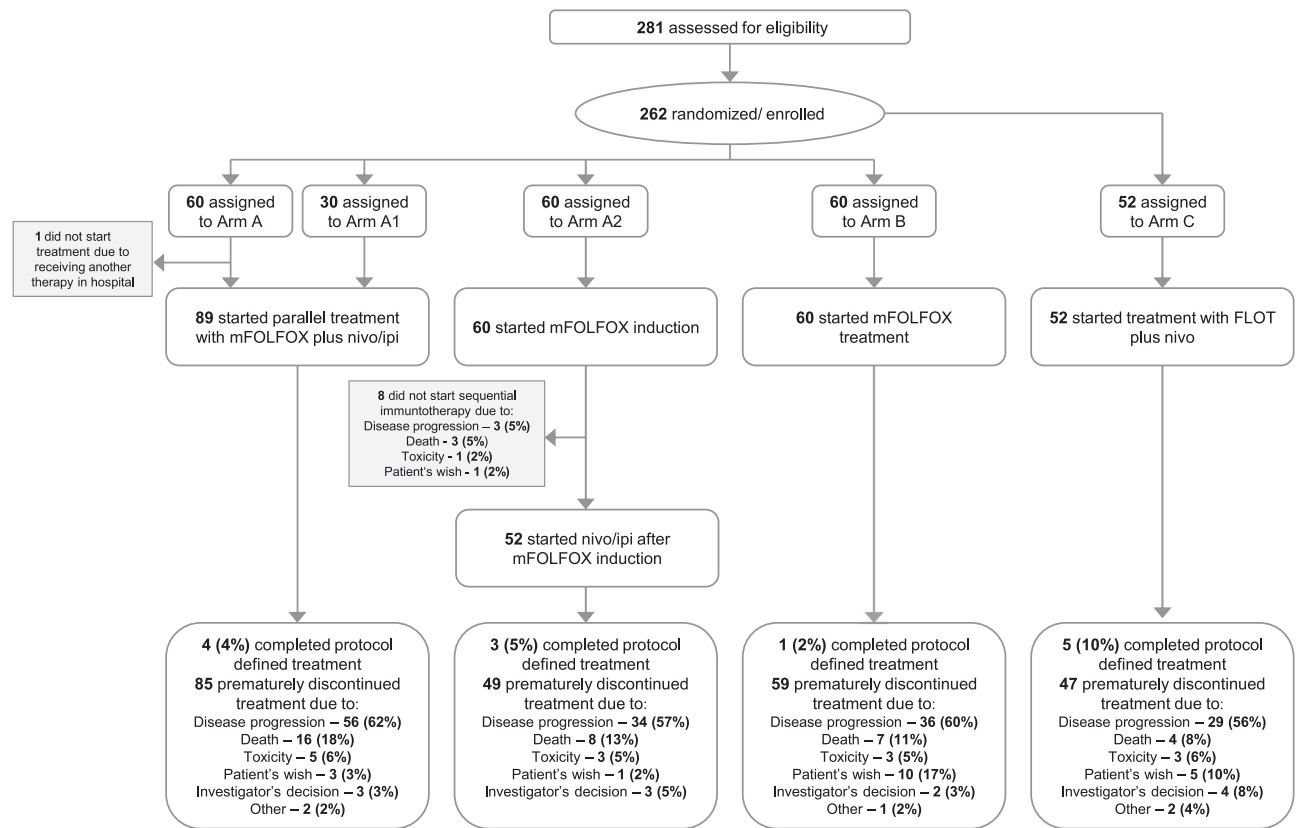

**Fig. 1 | CONSORT diagram.** All allocated patients were included into the intention-to-treat (ITT) analysis sets. The safety population comprised all patients who received at least one dose of study treatment. Since the only difference between Arm A and A1 was the study period in which the patients were enrolled, the data of both arms were pooled for explorative analyses. Ipi ipilimumab, FLOT (5-FU, leucovorin, oxaliplatin, docetaxel), mFOLFOX (5-FU, leucovorin, oxaliplatin), nivo nivolumab.

during or directly after chemotherapy induction due to progression, death, unacceptable toxicity or patient's request and did not start the following immunotherapy. In Arm A/A1 the median number of treatment cycles was 10 versus 8 in Arm A2 and the median number of nivolumab/ipilimumab cycles was 10/3 and 4/2 for Arm A/A1 and A2, respectively. Of note, the median number of oxaliplatin/5FU cycles was 7/10 vs. 3/3 for Arm A/A1 vs A2. 23 patients (38%) received chemotherapy re-induction in Arm A2. Median number of treatment cycles in Arm B was 9. Further information about treatment exposure is available in the appendix (Table S1).

### Triplet chemotherapy plus IO (Arm C)
The median treatment duration in patients receiving FLOT in combination with nivolumab (Arm C) was 4.4 months and 5 patients (10%) completed the maximum treatment duration of 24 months. The most common reasons for treatment discontinuation were disease progression (56%) followed by patient's wish (10%). Three patients (6%) discontinued treatment due to unacceptable toxicity (Fig. 1). The median number of completed therapy cycles was 9.5.

### Efficacy
**Nivolumab plus ipilimumab plus chemotherapy versus chemotherapy (Arm A/A1 vs B).** The addition of doublet immunotherapy to mFOLFOX treatment did not result in improvement of clinical outcome of the patients. Patients in Arm A/A1 and Arm B showed comparable results regarding the clinical outcomes PFS rate at 6 months (48% [95% CI 37;59] vs. 47% [95% CI 34;60]), median PFS (5.8 months vs. 6.6 months; 95%CI 5.4;7.6) and median OS (10.1 months vs. 12.5 months) (Fig. 2). The median follow-up time (range) of the patients was 9.9 months (0.8–48.1) and 10.8 months (1.1-19.4). Furthermore, the ORR in both arms was 46% and 47%, with six patients in

Arm A/A1 and three patients in Arm B who achieved complete response (Table 2).

Nivolumab and ipilimumab plus chemotherapy did not improve survival compared to chemotherapy alone across most PD-L1 expression subgroups. In patients with PD-L1 combined positivity score (CPS) ≥1 median PFS for arms A/A1 ($n = 42$) vs B ($n = 23$) was 5.4 and 6.1 months, median OS was 9.2 vs. 12.5 months and median duration of response was 7.7 vs. 5.1 months. Similarly, in patients with PD-L1 CPS < 1 for arms A/A1 ($n = 36$) and B ($n = 24$), median PFS (6.5 vs 5.6;) and median OS (9.9 vs. 8.4) was comparable between treatment arms (Fig. S1).

### Nivolumab plus ipilimumab plus chemotherapy versus alternating approach (Arm A/A1 vs A2)
The parallel combination of immunotherapy with mFOLFOX (Arm A/A1) was compared with an alternating approach (Arm A2). Here, the PFS rate at 6 months was significantly higher in patients receiving parallel treatment compared to those receiving the sequential treatment (48% [95% CI 37;59] vs. 30% [95% CI 19;43], $p = 0.041$). Also the median PFS (5.8 months vs. 4.0 months; (95%CI 3.6;5.5) and OS (10.1 months vs. 7.6 months) were prolonged in Arm A/A1 compared to Arm A2, in which the median follow-up time was 7.5 months (1.4–45.0) (Fig. 2). Also ORR was increased in Arm A/A1 (46%) compared to Arm A2 (32%) (Table 2). Prespecified subgroup analysis for PD-L1 expression showed longer median PFS and OS in Arm A/A1 compared to A2, independent of the CPS cut-off <1 or ≥1. The Kaplan–Meier plots for PFS and OS are shown in the appendix (Supplementary Fig. S2).

### FLOT plus nivolumab (Arm C)
In Arm C, evaluating patients treated with FLOT in combination with nivolumab, the PFS rate at 6 months was 46% [95% CI 32;61], the median PFS was 7.0 months [95% CI 5.2;9.5] and the median OS was

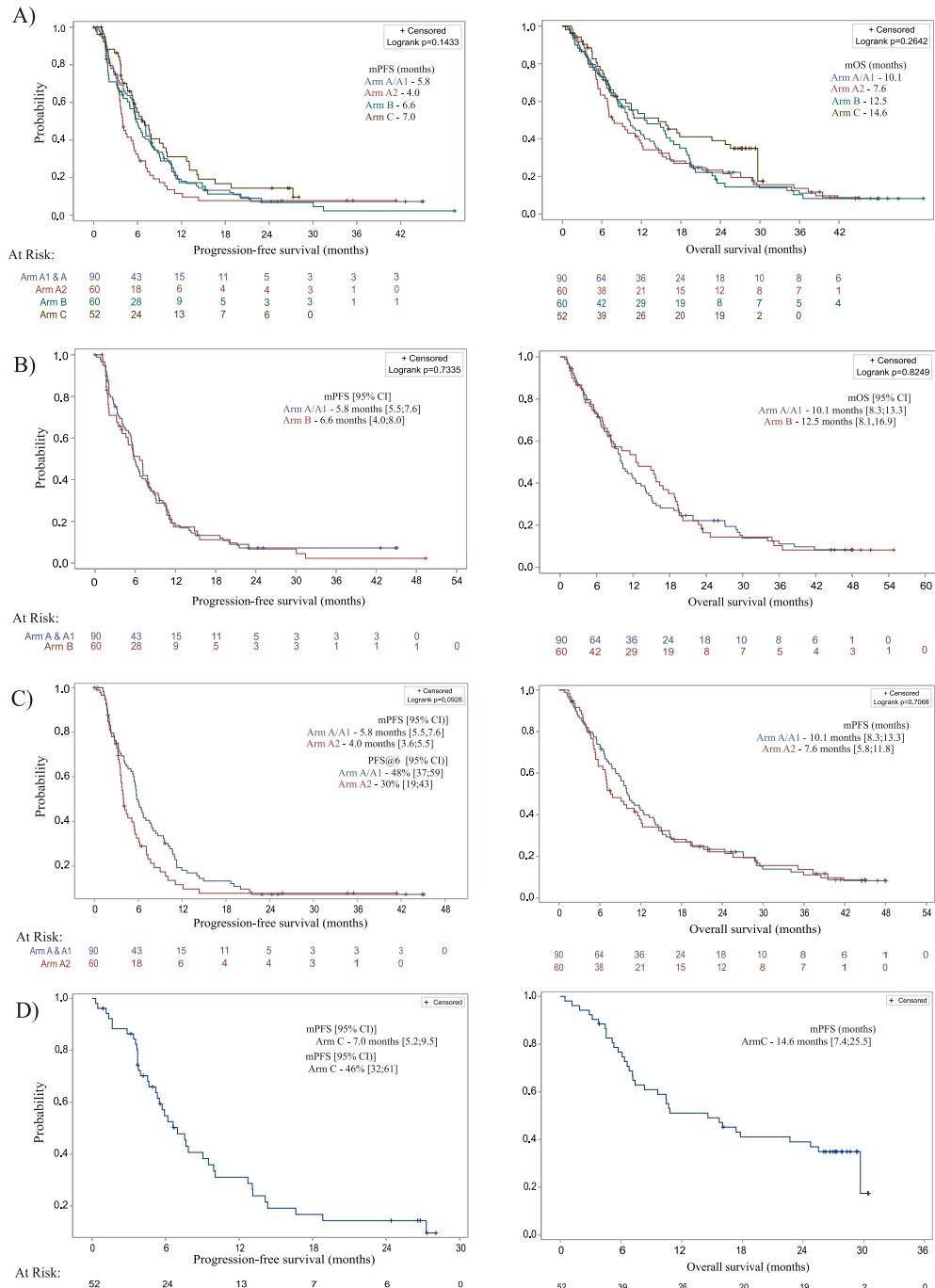

**Fig. 2 | Progression-free and overall survival. A** Kaplan-Meier estimates of progression-free and overall survival in patients treated with mFOLFOX plus nivolumab/ipilimumab in parallel (Arm A/A1) versus patients treated sequential with mFOLFOX plus nivolumab/ipilimumab (Arm A2) versus patients treated with mFOLFOX alone (Arm B) versus patients treated with FLOT (Arm C). Source data are provided as a Source data file. **B** Kaplan-Meier estimates of progression-free and overall survival in patients treated with mFOLFOX plus nivolumab/ipilimumab in parallel (Arm A/A1) versus patients treated with mFOLFOX alone (Arm B). Source data are provided as a Source data file. **C** Kaplan-Meier estimates of progression-free and overall survival in patients treated with mFOLFOX plus nivolumab/ipilimumab in parallel (ArmA/A1) versus patients treated sequential with mFOLFOX plus nivolumab/ipilimumab (Arm A2). Source data are provided as a Source data file. **D** Kaplan-Meier estimates of progression-free and overall survival in patients treatedwith FLOT plus nivolumab (Arm C). Source data are provided as a Source data file.

14.6 months [95% CI 7.4; 25.5], with a median follow-up time of 12.7 months (0.5–30.5). The ORR was 56%, where 5 (10%) patients achieved CR and 24 (46%) achieved PR (Fig. 2, Table 2). With respect to PD-L1 expression, median OS and PFS did change notably according to PD-L1 CPS < 1 ($n$ = 13) vs ≥1 ($n$ = 34). Median PFS was 3.9 months for CPS < 1 vs 6.6 months for CPS ≥ 1, as was median OS with 10.5 months vs 16.7 months, respectively (Supplementary Fig. S3).

## Subsequent therapy

Subsequent therapy was received by 59% and 63% of patients in the nivolumab plus ipilimumab plus chemotherapy arm (A/A1) and chemotherapy alone arm (B), and by 65% of patients in the alternating nivolumab-plus-ipilimumab and chemotherapy arm (A2). 52% of patients in the FLOT plus nivolumab arm received further treatment.

**Table 2 | Tumor response of the intention-to-treat population according to treatment arm**

| | Arm A/A1 N = 90 | Arm A2 N = 60 | Arm B N = 60 | Arm C N = 52 |
|---|---|---|---|---|
| Complete response | 6 (7%) | 6 (10%) | 3 (5%) | 5 (10%) |
| Partial response | 35 (39%) | 13 (22%) | 25 (42%) | 24 (46%) |
| Stable disease | 29 (32%) | 26 (43%) | 14 (23%) | 16 (31%) |
| Progressive disease | 11 (12%) | 8 (13%) | 12 (10%) | 3 (6%) |
| Missing | 9 (10%) | 7 (12%) | 6 (10%) | 4 (8%) |
| Objective response rate [95% CI] | 46% [35%,56%] | 32% [20%,45%] | 47% [34%,60%] | 56% [41%,70%] |
| Disease control rate | 78% | 75% | 70% | 87% |
| Median duration of response [95% CI] | 6.2 months [4.0;8.8] | 4.3 months [1.9;8.7] | 5.4 months [4.0;9.0] | 8.8 months [5.3; 12.6] |
| Median duration of stabilization [95% CI] | 4.9 months [3.8;7.7] | 3.5 months [2.0;5.2] | 6.9 months [4.8;9.0] | 5.7 months [3.4;8.1] |

## Safety

Almost all patients in Arm A/A1, A2 and B had at least one AE, whereas the incidence of AEs classified as treatment-related with grade ≥3 was increased in Arm A/A1 compared to Arm A2 and B (74% vs. 45% vs. 45%). In addition, more patients in Arm A/A1 had an SAE (75% vs. 62% vs. 50%) and specifically the incidence of treatment-related SAEs grade ≥3 was clearly increased in arm A/A1 (42% vs. 17% vs. 18%). In Arm C, all patients had at least one AE and 67% and 23% of patients had at least one treatment-related AE grade ≥3 and treatment-related SAE grade ≥3, respectively.

Most common AEs in all arms were anorexia, constipation, diarrhea, fatigue, nausea, peripheral sensory neuropathy and vomiting (Table S2). Treatment-related AEs grade ≥3 which were more prevalent in Arm A/A1 compared to Arm A2 and B were neutrophil count decrease (37% vs. 18% vs. 17%), platelet count decrease (5% vs. 0% vs. 0%), sepsis (8% vs. 2% vs. 0%) and white blood cell decrease (9% vs. 5% vs. 5%).

The number of fatal SAEs reported was 16 in Arm A/A1, 11 in Arm A2 and 9 in Arm B. Of these, 5, 1 and 3, respectively, were classified as treatment-related: In Arm A/A1 autoimmune hepatitis, Lyell-syndrome, pneumonitis, sepsis and deterioration of general conditions, in Arm A2 sepsis and in Arm B lung infection.3x Six fatal SAEs were reported for Arm C, with one of them (gastric perforation) classified as treatment-related (Table 3).

## Discussion

MOONLIGHT evolved into a platform trial of sequential, randomized and single-arm cohorts investigating the combination of dual immunotherapy with FOLFOX in parallel versus sequentially and versus chemotherapy alone, as well as triplet FLOT plus nivolumab.

Until now the relative contribution of adding the CTLA-4 checkpoint inhibitor ipilimumab to first-line combined nivolumab + chemotherapy for patients with advanced gastroesophageal adenocarcinoma has not been investigated. So far, also the triplet regimen FLOT in combination with nivolumab has not been evaluated.

Given the favourable response- and progression-free survival rates of the combination of anti-CTLA-4 and PD-L1 in the treatment of metastatic melanoma9 and in advanced chemo-refractory GEA,7 it seemed reasonable to compare the addition of nivolumab/ipilimumab to chemotherapy head-to-head with chemotherapy and in a sequential approach.

At the time MOONLIGHT was designed, data on the effectiveness of checkpoint inhibitors and their dependence on the level of PD-L1 expression were preliminary. Therefore, MOONLIGHT included patients independent of the PD-L1 expression status.

In MOONLIGHT, dual checkpoint inhibition plus Chemotherapy administered in parallel was associated with no improvement in activity compared to chemotherapy alone, with comparable survival and response rates, and there was no enrichment with increasing PD-L1 CPS cut-offs. Comparable results, with no additional benefit of adding the CTLA-4 checkpoint inhibitor Tremelimumab to FOLFIRI plus Durvalumab, independent of PD-L1 CPS in second-line GEA have been reported recently by the French DURIGAST Trial.[17] In MOONLIGHT, for chemo alone, the median OS and median PFS were even numerically longer and were in line with survival rates reported by the CheckMate 649[14] and KEYNOTE 859 trial[18] for the standard chemotherapy arm. One explanation for this observation may be the higher number of treatment-related adverse events of grade 3/4 which were seen with dual immunotherapy in combination with chemotherapy (TRAEs 74% vs. 45%). It is known that combining anti-PD-L1/anti-PD-1 with anti-CTLA-4 increases the proportion of grade 3 to 4 AEs.[7] Recently, published results also suggested an unfavorable benefit–risk profile with no improvement in survival but a higher number of treatment-related adverse events and deaths for nivolumab–ipilimumab combined with platinum-based chemotherapy relative to pembrolizumab combined with platinum-based chemotherapy as a first-line treatment for patients with advanced NSCLC.[19]

Another explanation for the failure of adding immunotherapy might be the generally low number of PD-L1 positive patients with less than 50% harboring a PD-L1 status of ≥1 (antibody clone 28-8), as trials in GEA have revealed that only the subset of patients with high-PD-L1 expression on tumor and immune cells substantially benefit from immune checkpoint inhibitors.[14,16,18] Compared to PD-L1 expression rates reported from pivotal global phase III trials in HER2 negative GEA, ranging between CPS ≥ 70–80[14,18] PD-L1 expression in European trials were indicated lower.[20,21] A meta-analysis by Lin et al.[22] found that the PD-L1 expression rate in studies from Asian areas was numerically higher (52.3%) compared to those from non-Asian areas, including Caucasians (32.7%), which might be due to higher frequencies of virus-associated malignancies in Asian areas. However, data are conflicting as trials in other malignancies such as lung cancer indicate, despite some regional differences, a relatively consistent PD-L1 expression across regions like Europe, Asia Pacific, America.

While there are some differences in PD-L1 expression rates between European and international studies, these differences may be also influenced by regional, methodological, and interlaboratory factors.

Of note, OS and PFS did not seem to vary according to PD-L1 CPS. The reason why we didn't observe a difference in the magnitude of survival benefit between patients with and with no PD-L1 expression might be the small number of patients, the generally low PD-L1 expression rate, and the chosen cut-off ≥1, as trials

**Table 3 | Summary of adverse events according to treatment arm in the safety population**

|  | Arm A/A1 N = 89 | Arm A2 N = 60 | Arm B N = 60 | Arm C N = 52 |
|---|---|---|---|---|
| Any AEs | 89 (100%) | 60 (100%) | 59 (98%) | 52 (100%) |
| Treatment-related AEs | 85 (96%) | 52 (87%) | 57 (95%) | 50 (96%) |
| Treatment-related AEs grade ≥ 3 | 66 (74%) | 27 (45%) | 27 (45%) | 35 (67%) |
| Treatment-related SAEs | 42 (47%) | 12 (20%) | 14 (23%) | 17 (33%) |
| Treatment-related SAEs grade ≥ 3 | 37 (42%) | 10 (17%) | 11 (18%) | 12 (23%) |
| Treatment-related fatal SAEs | 5 (6%) | 1 (2%) | 3 (5%) | 1 (2%) |

demonstrated that only the subset of patients with high PD-L1 expression on tumor and immune cells benefit substantially from immune checkpoint inhibitors.[14,16,18]

The second question, if an alternating regimen with immune maintenance therapy in patients with disease control after short term induction chemotherapy alone is as effective and less toxic as continuous chemoimmunotherapy showed a markedly longer OS with FOLFOX and nivo and ipi in parallel than that of FOLFOX induction followed by nivolumab and ipilimumab (16.5 months vs. 6.9 months) either in the overall or prespecified PD-L1–positive population (CPS ≥ 1%). Similar results have been reported in JAVELIN Gastric 100 trial, which failed to demonstrate superior OS with avelumab maintenance versus continued chemotherapy in patients with advanced GC or GEJ cancer.[20] Of note, Javelin had a longer chemotherapy induction of 3 months and did not show a detrimental effect for IO maintenance therapy. Of note, median time of chemotherapy treatment in the sequential treatment was only 6 weeks (3 cycles) with only 38% of patients receiving chemotherapy re-induction after immunotherapy, meaning that in a disease as aggressive in terms of tumor biology as GEA, prolonged chemotherapy is necessary to control the tumor. Although the number of subsequent therapies was above average what is reported from phase III trials and was highest in the sequential treatment arm (65%), the subsequent chemotherapy could not compensate the early progression on immune- maintenance therapy. Despite associated with lower toxicity and a clearly more favorable safety profile versus the combined chemotherapy plus IO therapy, including lower rates of grade ≥ 3 TRAEs (74% v 45%), reductions in treatment-related hematologic AEs, and a lower incidence of neuropathy, our study doesn't support the concept of chemotherapy induction followed by immunotherapy.

Third, MOONLIGHT addressed the questions of triplet FLOT regimen in combination with immunotherapy, which is of importance, as recently the French GASTFOX trial confirmed that the addition of docetaxel to oxaliplatin and 5-fluorouracil (mFLOT/TFOX) is more effective than oxaliplatin and 5-fluorouracil (FOLFOX) for patients with advanced gastric or gastroesophageal junction (GEJ) adenocarcinoma.[23] This regimen is considered a new first-line treatment option for patients eligible for triplet regimens, at least for patients with PD-L1-negative tumors according to current guidelines.[24] Although in label according to EMA approval, there have been no data on safety and efficacy of a triplet chemotherapy regimen in combination with immunotherapy so far. Data on the combination of immunotherapy with triplet taxan containing chemotherapy have been only conducted in a single arm phase II trial in locally advanced head and neck cancer, reporting favorable response and survival rates with good tolerance.[25] In MOONLIGHT, the combination of nivolumab and FLOT in 52 patients with metastatic GEA was associated with a favorable ORR and an improved disease control in the early course of therapy as well as a prolonged

survival rate. Besides, the safety profile of nivolumab plus FLOT chemotherapy was consistent with the known safety profiles of the individual treatments and no new safety signals were identified. Nivolumab plus chemotherapy-related grade three or worse events were 67% and the most common grade 3 AE was a neutrophil decrease in 44%. These data were consistent with the known safety profile of checkpoint inhibitors with doublet chemotherapy as shown in CheckMate 649[14] and Keynote 859[18] with similar frequencies of Grade 3-4 TRAEs (CM 649 in 60%; KN 859 59%).

Of note, baseline characteristics in this group were more favorable for the success of immuno-chemotherapy. By immunohistochemistry evaluation, PD-L1 expression degrees was highest in the FLOT plus nivolumab cohort with 65% harboring a PD-L1 CPS of ≥1. Both, Keynote 859 and CheckMate 649 determined a meaningful prolongation of survival treated with pembrolizumab/nivolumab in metastatic GEA patients with more than 1%/5% PD-L1 expression.[14,18] Besides, FLOT plus nivolumab led to a numerically longer median OS in patients with a tumor PD-L1 expression with a median OS of 16.7 months in PD-L1 positive patients vs 10.6 months in PD-L1 negative patients. Between trial comparisons of triplet FLOT plus nivolumab in MOONLIGHT vs doublette FOLFOX plus Pembrolizumab in Keynote 859 with a median OS of 13 months in CPS ≥ 1 patients, anticipates an increased efficacy with the addition of the taxane. These results suggest that adding nivolumab to FLOT is an effective treatment option with good tolerance and may represent an option for selected patients with aggressive, rapidly progressive disease. However, given the small cohort size and lack of control for evolving treatment standards, results must be interpreted with extreme caution. A potential next step could be to evaluate mFLOT/TFOX or FLOT regimen in combination with immunotherapy or zolbetuximab in biomarker selected patients.

Moonlight had several limitations. First, patients were recruited independent of biomarker selection based on PD-L1 score and mismatch repair deficiency, although rare, was not measured. Overall, PD-L1 expression rate was comparable between arms but generally low compared to previous reported trials and the cohort with the most promising results (FLOT plus nivo) had the highest number of PD-L1 positive patients.

Second, sample size of each cohort was small and according to the statistical design, cohorts could not be compared statistically, as this study was not designed to compare these arms. Therefore, it is also not possible to speculate advantage of one or the other combination treatment for specific study subgroups which clearly limits the meaningfulness of this trial. The current study's contribution is therefore exploratory rather than practice-changing

Another limitation is that the experimental arms (A1/A2) and A2 have not been compared to the current first-line treatment standard of doublet chemotherapy plus IO monotherapy, as this was not yet standard of care at the time the trial was designed.

In conclusion, the results of MOONLIGHT suggest, that in the first-line setting of metastatic GEA, the combined immune checkpoint blockade with chemotherapy is insufficient and toxic and should not be further explored in future studies.

In addition, brief duration of 6 weeks of FOLFOX followed by immunotherapy alone is inferior irrespective of PD-L1 expression. Future trials should evaluate immune maintenance therapy after longer induction with chemoimmunotherapy in PD-L1 selected patients.

Encouragingly, the combination of FLOT with nivolumab is feasible and appears to show good efficacy, further supporting the use of this regimen for selected patients who need triplet chemotherapy for early tumor control and might benefit from additional immunotherapy due to positive PD-L1 expression. However, the potential benefit of FLOT plus immunotherapy should also be carefully weighed against its notable toxicity.

## Methods

### Ethics statement

The MOONLIGHT trial was conducted in accordance with the Declaration of Helsinki and complies with all relevant ethical regulations. The trial protocol, including all subsequent amendments, was reviewed and approved by the Ethikkommission der Landesärztekammer Hessen (approval number: 2019-1234-fAM) as coordinating ethics committee. In addition, ethical approval was obtained from the respective ethics committees at each participating center prior to patient enrollment. All participants provided written informed consent before inclusion in the study. This trial is registered with ClinicalTrials.gov under identifier NCT03647969.

This trial was an investigator initiated, interventional, prospective, randomized, open label, multicenter phase II trial followed by a non-randomized arm. Participants were adults (≥18 years) of any sex and ethnicity. Patients self-reported their sex and ethnicity; no information on gender was collected. Eligible patients had previously untreated, histologically confirmed inoperable, advanced or metastatic, HER2-negative gastric or gastro-oesophagal junction (GEJ) adenocarcinoma, an Eastern Cooperative Oncology Group (ECOG) performance status ≤1 and adequate haematological, hepatic and renal function parameters. Inclusion was regardless of PD-L1 expression- and microsatellite instability (MSI) status.

PD-L1 expression on viable tumor cells was assessed centrally using a laboratory developed immunohistochemical PD-L1 staining according to current guidelines using the primary antibody clone 28-8 (Ab 205921).

### Enrollment and randomization

Patients were randomly assigned in a 1:1 ratio into Arm A (mFOLFOX: Oxaliplatin 85 mg/m², leucovorin 400 mg/m² and fluorouracil 400 mg/m² plus Nivolumab and Ipilimumab) and B (mFOLFOX). In a subsequent phase, patients were randomized in a 1:2 ratio into Arm A1 (identical to Arm A) and A2 (three cycles of induction chemotherapy with mFOLFOX followed by four administrations of nivolumab (Q2W) and two administrations of ipilimumab). Afterwards enrollment into Arm C (FLOT: Docetaxel 50 mg/m², Oxaliplatin 85 mg/m², leucovorin 200 mg/m² and fluorouracil 2600 mg/m² plus Nivolumab) was performed in a single arm design. (For details, see the Supplementary Information)

### Study objective and endpoints

The primary objective was to determine the clinical performance of the experimental regimen. The primary endpoint was progression free survival according to RECIST v1.1 in Arm A vs. Arm B and PFS rate at 6 months (PFS@6) in Arm A2 and Arm C. Secondary objectives were to determine efficacy in terms of objective response rate (ORR) according to RECIST v1.1, overall survival (OS), PFS for Arm A1, A2 and C, PFS@6 for Arm A and Arm B, PFS and OS by PD-L1 expression status, as well as tolerability of the experimental regimen and Quality of Life.

### Statistical analysis

Unless otherwise stated, the results were based on the intention-to-treat (ITT) population, which comprised all randomized or enrolled (Arm C) patients in the study. Since the only difference between Arm A and A1 was the study period in which the patients were enrolled, the data of both arms were pooled for explorative analyses if the baseline characteristics are balanced. Therefore, a pooled Arm A/A1 was used for all endpoints in comparison with B and A2, respectively. The results of Arm C were compared descriptively with the results of Arm A and B. More details on sample size calculation are provided in the data Supplementary Information section.

The primary endpoint PFS was estimated using Kaplan Meier techniques. Median PFS along with 95% confidence interval (CI) and

hazard ratio (HR) for PFS along with 95% CI was calculated and the hypothesis testing was performed with a one-sided log-rank test at a one-sided significance level of 0.1.

Except for the primary endpoint, all parameters were evaluated in an explorative or descriptive manner, providing means, medians, interquartile and total ranges, standard deviations and/or confidence intervals, counts and proportions, or Kaplan-Meier curves, as appropriate for the respective data types.

### Reporting summary

Further information on research design is available in the Nature Portfolio Reporting Summary linked to this article.

## Data availability

All requests for data will be reviewed by the leading clinical site, Klinikum rechts der Isar, TUM University Rechts der Isar Hospital Munich, to verify whether the request is subject to any intellectual property or confidentiality obligations. Requests for access to the patient-level data from this study can be submitted via email to sylvie.lorenzen@mri.tum.de with detailed proposals for approval and will be responded to in two weeks. A signed data access agreement with the sponsor is required before accessing shared data. The study protocol is included within the Supplementary Information. Source data are provided with this paper.

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

## Acknowledgements

We thank the investigators who participated in this study and who have not been included in the co-author list of this publication. We warmly thank all IKF Headquarters team members who have not been included in the co-author list of this publication and who contributed to the study's conduct. Finally, we wish to thank the patients and their families for participating in this study and providing their invaluable contribution.

## Author contributions

S.L., T.O.G. and S.E.A-B contributed to the conception and design of the study in collaboration with Bristol Myers Squibb. S.L., T.O.G., P.C.T-P., J.R-K., E.G., T.N.D., T.J.E., R-D.H., K.B.L., D.P., U.L., G.F., G.S., M.B., V.H., S.A., C.B. and S.E.A-B. recruited and/or treated patients and M.L. conducted statistical analyses. S.L., T.O.G., S.J., C.P. and S.E.A-B verified the data. All authors interpreted the data. All authors had access to all the data in the study, participated in developing or reviewing the paper and provided final approval to submit the paper for publication.

## Funding

 The sponsor of the trial is the Frankfurt Institute of Clinical Cancer Research IKF GmbH. Financial support and drug supply for the conduct of the trial was granted by Bristol-Myers Squibb GmbH & Co. KGaA.

## Competing interests

The authors have no conflicts of interest to declare. I declare the authors have no competing interests as defined by Nature Portfolio, or other interests that might be perceived to influence the interpretation of the article. All co-authors have seen and agree with the contents of the manuscript and there is no financial interest to report.

## Additional information

¹TUM University Hospital, Technical University of Munich, Department of Medicine III, Munich, Germany. ²The Frankfurt Institute of Clinical Cancer Research and Krankenhaus Nordwest, University Cancer Center Frankfurt, Frankfurt, Germany. ³Charité–Universitätsmedizin Berlin, Medizinische Klinik mit Schwerpunkt Hämatologie, Onkologie und Tumorimmunologie, Berlin, Germany. ⁴Universitätsklinikum Marburg, Klinik für Innere Medizin, Marburg, Germany. ⁵Hämatologisch-Onkologische Praxis Eppendorf (HOPE) und Universitäres Cancer Center Hamburg (UCCH), Hamburg, Germany. ⁶Onkologie Ravensburg, Ravensburg, Germany. ⁷Ulm University Hospital, Department of Internal Medicine I, Ulm, Germany. ⁸Universitätsmedizin Mannheim, Tagestherapiezentrum am ITM, Mannheim, Germany. ⁹University Hospital Schleswig-Holstein, Campus Luebeck, Lübeck, Germany. ¹⁰Klinik und Poliklinik für Innere Medizin C,

Hämatologie und Onkologie, Transplantationszentrum, Palliativmedizin, Universität Greifswald and Klinik für Hämatologie, Onkologie und Palliativmedizin-Sarkomzentrum, HELIOS Klinikum Bad Saarow, Bad, Saarow, Germany. [11]Universitätsklinikum Jena, Klinik für Innere Medizin II, Jena, Germany. [12]Universitätsklinikum Carl Gustav Carus, Medizinische Klinik I, Dresden, Germany. [13]Hämatologisch-Onkologische Praxis Altona (HOPA), Hamburg, Germany. [14]Universitätsklinikum Tübingen, Medizinische Klinik I, Tuebingen, Germany. [15]Klinikum der Universität München-Großhadern, Medizinische Klinik III, München, Germany. [16]Klinikum Ludwigsburg, Klinik für Innere Medizin, Gastroenterologie, Hämato-Onkologie, Pneumologie, Diabetologie und Infektiologie, Ludwigsburg, Germany. [17]AMEOS Rehaklinikum Ratzeburg, Germany and Agaplesion Markus Krankenhaus, Hämatologie/Onkologie, Frankfurt, Germany. [18]The Frankfurt Institute of Clinical Cancer Research, Frankfurt, Germany. ✉e-mail: sylvie.lorenzen@mri.tum.de

