## [Transparent Peer Review file · Nature Communications]

First-line modified FOLFOX plus/minus Nivolumab and Ipilimumab or FLOT plus Nivolumab in advanced gastroesophageal adenocarcinoma: a phase II multi-cohort trial

Corresponding Author: Professor Sylvie Lorenzen

Version 1:

Reviewer comments:

Reviewer #1

(Remarks to the Author)

Thank you for allowing me to reevaluate the manuscript. The authors have responded adequately to my questions and I have no further comments.

Reviewer #4

(Remarks to the Author)

This multi-cohort study explores several novel first-line treatment strategies for advanced gastroesophageal adenocarcinoma. The manuscript is clearly written and the core findings—that parallel mFOLFOX+nivolumab+ipilimumab increases toxicity without improving efficacy, that short FOLFOX induction followed by dual IO underperforms, and that FLOT+nivolumab is feasible with encouraging activity—are important for the field. The statistical analysis methods are generally appropriate for the study design, but several key areas require clarification and more rigorous presentation.

1. What are the key patient characteristics? Are they age, sex, ECOG performance etc.? While Table 1 provides descriptive statistics, it should be properly stated in the method section.
2. Are the covariates balanced between the arms (e.g., love plots, SMDs etc.). While the rationale for pooling Arm A and A1 is explained, authors should include a brief sentence in the results/supplementary section confirming that a comparison of baseline characteristics between these two arms did not reveal any significant differences that would preclude pooling.

This is a valuable exploratory study that addresses important questions in the treatment of gastroesophageal adenocarcinoma. The authors have correctly identified many of the trial's limitations. However, a more rigorous presentation of the statistical findings, particularly regarding covariate balance. The paper should be accepted for publication after a minor revision addressing the points raised above.

Reviewer #5

(Remarks to the Author)

I had pointed out a number of limitations of this study that do not allow one to make any efficacy conclusions (only safety data may be reliable), however, the authors only agreed to my comments and did not make any satisfactory revisions or responses.

Discussing some items in the Discussion section does not solve the problems pointed.

Scientific rigor is lacking.

Reviewer #6

(Remarks to the Author)

In this revised paper, Lorenzen and colleagues have addressed the majority of the concerns, particularly the study limitations, however a few issues remain unaddressed.

1. For completeness, the TNM stages of the participants should be addressed in the text under methods and not just in the abstract
2. I dont like the term "aggressive tumor biology" what does this mean? It should be removed

Point-by-point reply to the reviewers' comments:

Reviewer 1:

Summary of Key Results:

This multi-cohort trial explores the feasibility and preliminary efficacy of combining immunotherapy (nivolumab ± ipilimumab) with chemotherapy (modified FOLFOX or FLOT) in first-line treatment of gastric and gastroesophageal junction adenocarcinoma. While the study offers relevant early signals, several limitations—especially regarding toxicity and subgroup interpretation—should be addressed before drawing broad conclusions

AU: Agree. The limitations are now pointed out in the key results section.

Originality and Significance:

The study investigates an important clinical question in the evolving treatment landscape. While the idea of chemoimmunotherapy combinations is not novel, this trial adds value through its exploration of sequencing and regimen choice. However, it should be noted that the current study's contribution is exploratory rather than practice-changing

AU: Agree. We included this statement in the discussion section

Data and Methodology:

The overall study design is appropriate for a signal-seeking phase II trial. However, clarity is needed on key inclusion criteria such as PD-L1 CPS and MSI status. The rationale behind treatment allocation—particularly the use of FOLFOX alone in one arm—requires further justification, especially considering evolving standards of care during the study period. Furthermore, it is important to clarify whether patients received complete FLOT regimens or modified versions approximating FOLFOX.

AU: Agree. Inclusion criteria were specified in the Material and Methods section. This is also outlined in the discussion section: „At the time MOONLIGHT was designed, data on the effectiveness of checkpoint inhibitors and their dependence on the level of PD-L1 expression were preliminary. Therefore MOONLIGHT included patients independent of the PD-L1 expression status.

According to the protocol patients received FLOT regimen and no modified versions.

Statistical Approach and Uncertainty:

While the manuscript adequately reports descriptive data, the small sample size limits the robustness of subgroup analyses. In particular, conclusions related to PD-L1 positivity (e.g., CPS ≥ 1 or ≥ 5) must be tempered. Use of statistical tests appears appropriate, but uncertainties should be discussed more transparently, especially when interpreting trends without formal hypothesis testing.

AU: According to the protocol subgroup analysis including PFS and OS by PD-L1 expression status were planned but cut-offs were not defined as CM649 results were not available at the time of trial conception. Due to small number of patients in the prespecified cohorts, subgroup analysis on different CPS- cut-offs were not performed.

In addition, results are now discussed more cautious and in line of a signal-seeking phase II trial

Conclusions:

The conclusions should be more cautiously phrased, particularly regarding treatment benefit in patients with PD-L1 positivity or aggressive disease. Given the small cohort sizes and lack of control for evolving treatment standards, these claims are not yet supported by sufficiently robust evidence. The potential benefit of FLOT plus immunotherapy should also be more carefully weighed against its notable toxicity.

AU: agree. The discussion section was re-written accordingly.

Suggested Improvements:

- Provide consistent data for patients with CPS ≥ 5 across analyses.

AU: explained- too small subgroups

- Clarify if and how patients with “aggressive disease” can be identified at baseline.

AU: sentence was revised to “aggressive tumorbiology”

- Reconsider or revise the statement that FLOT-treated patients may benefit from additional immunotherapy based on PD-L1 status.

AU: done

- Include an acknowledgment in the Discussion regarding the relatively young age of the study population, which may limit generalizability.

AU: we agree that the median age in the trial population is younger compared to real-world setting, however, except for Arm A (58y) in the range of what is reported in other global registration trials (61-63y) such as CM649 (median Age 61-63y), and KN 859 (median Age 61y), and RATIONAL 305 (median Age 60y).

- Address minor phrasing and grammar corrections as noted, especially regarding clarity of the PD-L1 comparison with prior trials.

AU: done

Reviewer 3:

The following points should be addressed when revising the manuscript:

1. In the abstract, all abbreviations should be introduced before they are used (FLOT, FOLFOX).

AU: Due to the limited number of characters and the well established chemo regimens, we would refrain from explaining the abbreviations FLOT and FOLFOX in the abstract but explained on the body text (see point 3). Of course this can be implemented in the final version

2. In the abstract, the tumor stages investigated in the study (UICC TNM Stages) should be listed under "Patients and Methods".

AU: Stage was specified in the abstract in the background section

3. In the body of the text, all abbreviations should be introduced before they are used (FLOT, FOLFOX; GEA, EGA, HER2, CI, HR, GEJ...).

AU: done. GEA and EGA mean the same and have been standardized under GEA. Details on chemotherapy are also given in the „supplementary methods“ section

4. In the body of the text and in table 1, the tumor stages investigated in the study (UICC TNM Stages) should be given under "Patients and Methods".

AU: The trial was conducted exclusively in the metastatic stage, therefore the UICC/TNM stage is not applicable

5. The introductory section of the manuscript is very detailed and should be shortened.

AU: done

6. The discussion should include an explanation of why FOLFOX was used initially as the backbone of chemotherapy. Why wasn't FLOT used initially?

AU: done- this is also explain in the introduction. Triplet therapy with FLOT is not the recommended First-line therapy according to national and international guidelines

Reviewer 4:

The manuscript addresses an important clinical question in the treatment of HER2-negative gastroesophageal adenocarcinoma (GEA), evaluating innovative combinations of immunotherapy and chemotherapy. The study design and rationale are relevant, and the multi-cohort approach allows comparison of different treatment strategies. However, I recommend the following revisions to enhance clarity and accessibility:

1. Some abbreviations in the abstract, such as "pts" and "q2w," are not immediately clear. Please define all abbreviations upon first use in the main text, including commonly used terms like OS, PFS,

AE, and SAE, even if they appear in the abstract. Abbreviations should be introduced once and then used consistently throughout the manuscript.

AU: done. See also reviewer 3 point 3

2. Please add 95%CI for PFS in the abstract.

AU: Due to the limited number of words in the abstract (150 max), we decided to not include 95% CIs in the abstract but included it in the results section.

3. For broader accessibility, consider briefly explaining key terms such as immune checkpoint blockade, HER2-negative, and the distinction between triplet and doublet chemotherapy. These terms, though common in oncology, may not be immediately clear to non-specialist readers. A concise clarification would enhance readability for a broader biomedical audience.

AU: thanks- we address this important point - it was already raised by previous reviewers

4. In the phrase “NCTNCT03647969,” the prefix “NCT” appears twice. Please correct this to “NCT03647969.”

AU: done

5. The introduction is relevant and informative, but the narrative would benefit from improved structure. Currently, it moves quickly between themes such as prior trial results, biological rationale, treatment limitations, and the design for the MOONLIGHT trial, without clear paragraph breaks. Restructuring the introduction into logical sections would improve clarity and better convey the study’s significance.

AU: done

6. Ensure consistent formatting of figure and table references throughout the manuscript. For example, use capitalized “Figure” and “Table” rather than alternating between “figure” and “Figure.”

AU: done

7. When reporting estimates such as PFS rates, median PFS, or OS in the Results section, it would enhance clarity to include the confidence intervals in standard notation—for example, “46% [95% CI, 32–61]” rather than “46% [32;61].”

AU: done

8. I suggest - consider including a Restricted Mean Survival Time (RMST) causing a truncation time point (e.g., 12 months or 18 months). This can provide a more comprehensive summary of survival differences. (Ref: Zhao, Lihui, Brian Claggett, Lu Tian, Hajime Uno, Marc A. Pfeffer, Scott D. Solomon, Lorenzo Trippa, and Lee-Jen Wei. "On the restricted mean survival time curve in survival analysis." *Biometrics* 72, no. 1 (2016): 215-221.)

AU: In this study, we prioritized traditional survival analysis methods—specifically Kaplan-Meier estimates (with median survival times), the log-rank test, and hazard ratios—as they provide a comprehensive, clinically interpretable, and widely understood summary of survival differences. These methods preserve the full survival curve, thereby retaining long-term information without requiring an arbitrary truncation point, as would be necessary with RMST.

Additionally, RMST was not pre-specified in the Statistical Analysis Plan and thus was not included in the primary or secondary analyses

Reviewer 5:

As acknowledged in the manuscript, randomized and non-randomized arms with very limited number of patients creates considerable restrictions in figuring out results. This is starkly clear in various figures.

It would be inappropriate to combine non-randomized arm (C) with other groups.

sample size is too small and does not conform with PDL-1 pos/neg dataset on thousands of patients that already exists.

Any conclusions other than safety will have considerable weakness.

Rationale to conduct such a confusing research design is entirely unclear and not justifiable.

Arm A2: very confusing. short course of FOLFOX (??) and IO duration left to the investigators. very confusing and weak plan.

MS status not mentioned.

AU: We fully agree with the reviewer's arguments. We are aware of the limitations of the study design and have presented and discussed this openly and correctly. Nevertheless, we consider the signals emerging from the results to be important for the future use of immunotherapies in the first-line treatment of Her2 negative gastroesophageal adenocarcinoma, e.g. when it comes to discontinuing chemotherapy too early or combining it with triple chemotherapies such as FLOT or TFOX.